# Heart Rate Variability, Time Estimation and Internet-Dependent Behaviour in 16–17-Year-Old Adolescents: A Study in Russian Arctic

**DOI:** 10.3390/life11060497

**Published:** 2021-05-29

**Authors:** Olga Krivonogova, Elena Krivonogova, Liliya Poskotinova

**Affiliations:** N. Laverov Federal Center for Integrated Arctic Research of the Ural Branch of the Russian Academy of Sciences, 163069 Arkhangelsk, Russia; ja.olga1@gmail.com (O.K.); elena200280@mail.ru (E.K.)

**Keywords:** internet addiction, adolescents, time estimation, heart rate variability

## Abstract

Internet-dependent behaviour in adolescents can contribute to a change in the function of the nervous system, which is reflected in the violation of time perception and autonomic regulation of the heart rate. The aim of the study was to determine groups of individuals with different risks of Internet addiction (IA) in relation to heart rate variability (HRV) parameters and the efficiency of time estimation in adolescents aged 16–17 years living in the Russian Arctic. Adolescents aged 16–17 years (*n* = 49–32 females, 17 males) living in Yamalo-Nenets Autonomous Okrug (Russia) were observed. Chen Scale Internet Addiction (CIAS) was used. The duration of an individual 1 min was determined. HRV parameters were determined using the "Varicard" equipment (Russia). In 16–17-year-old adolescents with different levels of risk of developing IA, including signs of IA, we revealed a high severity of symptoms of withdrawal from Internet use, difficulty in time estimation against the background of sympathicotonia and a decrease in vagal regulation of heart rate. In individuals with minimal symptoms of withdrawal from Internet use, the total HRV and vagal activity remain higher than in those with severe withdrawal symptoms, and their time estimation remains effective.

## 1. Introduction

Internet overuse has led to the formation of Internet-dependent behaviour, which develops against a background of psychoemotional disorders and the development of a negative perception of the prospects for one’s personal development [1]. Internet-dependent behaviour, including a stable pattern of Internet addiction (IA), can significantly transform the health indicators of adolescents because in this age period there is an age-related formation of endocrine system and functional interneuronal connections in their cerebral cortex [2]. Heart rate variability (HRV) is under the direct control of the central and autonomic nervous systems and is one of the most important indicators of adaptive processes in the human organism [3]. A healthy person has a high HRV, which reflects a readiness to respond to environmental stimuli. A decrease in HRV indicates the possibility of pathological conditions, including cardiovascular diseases, and is detected in people who consume alcohol, tobacco, with prolonged exposure to electromagnetic fields, with fatigue, depression, anxiety, with the experience of negative emotions [4]. The predominance of the sympathetic nervous system’s activation has been observed in individuals with Internet overuse [5]. Lin et al. found that school-aged children with IA had lower HRV than children without IA [6]. However, there is information to show that individuals at high risk of developing IA have both increased sympathetic activity and activation of the parasympathetic nervous system [5]. The ability to accurately determine the duration of human-controlled events is important for human adaptation to the environment and self-control of behaviour [7,8]. Young people may lose control of time with prolonged use of the Internet [9]. However, it is not known at what level of risk of developing IA the loss of time control in combination with a decrease in vagal control of cardiac activity occurs. The high reactivity of an adolescent’s functional systems to information load is also combined with the impact of uncomfortable natural climatic conditions. The author’s use of different tests to determine IA and the participation of different age groups of young people allow us to approximately compare the physiological correlates of IA among adolescents in different countries. However, the spread of IA among young people significantly affects the socio-economic indicators of these countries [10]. For example, as of 2018, Chinese and Japanese students showed a higher risk of IA when compared to USA students [11]. Among Finnish adolescents, 24.2% of Internet users had excessive use of Internet resources and admitted that excessive Internet use causes mental, social, and physical harm in them [12]. In Norway, as of 2004, an average of 1.98% of young people have signs of IA, while up to 21.68% of young people who are active Internet users admitted to having psychosocial problems associated with its use [13]. As of 2020, the average values according to the Chen Internet Addiction Scale (CIAS), which reflects the general level of severity of Internet-addicted behaviour, in adolescents aged 16–17 years living in the northern and southern Russian regions were statistically the same. However, the symptoms of a stable IA pattern were somewhat more common in people living in the northern region (15.8%) than in the southern region (8.9%) of Russia [14]. The Russian Federation includes vast Arctic territories, whose residents experience constant stress on the mechanisms of adaptation of the central and autonomic nervous systems. Migrants from the southern regions of Russia in the Arctic show symptoms of sympathicotonia [15] and an increased level of psychoemotional stress [16]. It is therefore important to determine what IA symptoms cause a decrease in effectiveness of time estimation accompanied by impaired autonomic heart rate regulation in adolescents living in the Arctic. Consequently, the aim of this study was to determine groups of individuals with different risks of IA in relation to HRV parameters and the efficiency of time estimation in adolescents living in the Russian Arctic.

## 2. Materials and Methods

The participants in the non-randomized cross-sectional study were 49 apparently healthy adolescents aged 16–17 (32 girls, 17 boys, Europeoid residents), who were natives of Nadym (Yamalo-Nenets Autonomous Okrug of the Russian Federation—65°32″ N, 72°31″ E). This administrative region is part of the Arctic zone of the Russian Federation [17]. The participants were initially formed as a group of healthy individuals according to medical criteria based on the opinion of a pediatrician, as reflected in the personal medical documentation in the school’s medical office. The parents of most of the surveyed schoolchildren (90%) were descendants of migrants from the southwestern regions of the Russian Federation, and only 10% of the parents of the participants were natives of this region. All of the participants successfully performed the educational school programme. In addition, all of the school’s students could cope with the school curriculum, and none of them studied according to individual programmes due to physical or mental health disabilities. None of the participants smoked or consumed alcoholic drinks. Insomnia as a medical diagnosis was not verified in the study participants. The study participants had not committed any offences and did not need police or psychological supervision in connection with the manifestation of aggression, depression, and suicidal behaviour or the use of psychoactive substances. Anthropometric parameters (height and body mass index (BMI)) were measured for each participant in the school medical office. BMI was defined as the body mass divided by the square of the body height (kg/m^2^), with mass in kilograms and height in meters. Height, weight, and BMI were measured using certified medical equipment with electronic scales and a height meter ("REP+VMEN-200–100-D1-A", Russian Federation). The adolescents were not underweight or overweight (BMI 18.5–24.9 kg/m^2^). Furthermore, all of this study’s participants had constant access to the Internet through their personal smartphones and personal computers. The total time for using the Internet (for the purpose of learning in the school curriculum and for entertainment) averaged 2.5 h for 8.5% of individuals, 5.5 h for 42.9% of individuals, and 6 or more hours for 48.6% of individuals every day. The entertainment time included social media time, watching video content and news, and online games. For the purpose of entertainment, 15 people (42.9%) used the Internet 3–4 h a day, 19 people (54.3%) used the Internet 2–3 h a day, and 1 person (2.8%) used the Internet up to 2 h a day. For the purpose of entertainment, 15 people (42.9%) used the Internet more than 3 h a day, 19 people (54.3%) used the Internet 2–3 h a day, and 1 person (2.8%) used the Internet up to 2 h a day.

Internet-dependent behaviour was assessed using the Chen Internet Addiction Scale [18] in the Russian version that was developed by Malygin and Feklisov [19]. When studying Internet-dependent behavior in Russian people, the reliability measures of the CIAS test (Cronbach’s alpha) ranged from 0.757 to 0.9, depending on the CIAS subscales ([19], pp. 27–28). In other words, the CIAS test is presented with a sufficient degree of reliability. Symptoms of IA included compulsive symptoms (Com), withdrawal (Wit), tolerance (Tol), interpersonal and health-related problems (IH-RP), and time management (Tm) problems. The CIAS comprised a questionnaire with 26 items and a four-point Likert scale, ranging from 1 point (Does not match my experience) to 4 points (Definitely matches my experience). Thus, the minimum CIAS value was 26, and the maximum value was 104. Respondents with CIAS scores above 64 points were considered to have a stable pattern of Internet addiction, scores of 43 to 64 were associated with a moderate risk of developing IA, and those less than 43 points indicated a minimal risk of developing IA. The preliminary analysis did not reveal statistically significant differences between the CIAS scores between males and females; therefore, all of the indicators are presented in the general sample. 

The effectiveness of time estimation was determined by the duration of an individual minute (IM) [20]. When assessing the duration of IM, the researcher verbally designated the participant to start (the word was “Start!”). The participant then mentally counted seconds from 1 to 60. Simultaneously, the researcher used a stopwatch to count the seconds of physical (real) time. When the participant had counted 60 s, they verbally indicated the end of the counting (the word was “Sixty!”), and the researcher stopped the stopwatch. The real-time value was then estimated as the participant’s IM time in seconds. 

Heart rate variability was assessed using the "Varicard" equipment (Russian Federation). HRV indices were recorded in a person in a sitting position for 5 min. To characterize the heart rate (HR), we used the average heart rate for 5 min recording cardiointervalogram. The HRV indicators that we used included heart rate (HR, bpm), standard deviation of all NN (cardio intervals) intervals (SDNN, ms), the square root of the mean squared differences of successive NN intervals (RMSSD, ms), the percentage of the number of pairs of consecutive cardio intervals differing by more than 50 ms (pNN50, %), and Stress Index (SI, units). SI was calculated by the formula (SI = Amo50/2 × VAR × Mo, where Mo (ms) is the cardiointerval value dividing the cardio-interval-gram series in half, VAR is the variation range between the minimum and maximum values in the cardio-interval-gram series, and AMo50% (Mode amplitude) is the number of cardiointervals); total power of HRV spectrum (TP, ms^2^); high frequency range of HRV spectrum, 0.15–0.4 Hz (HF, ms^2^); low frequency range of HRV spectrum, 0.04–0.15 Hz (LF, ms^2^); very low frequency range of HRV spectrum, 0.04–0.003 Hz (VLF, ms^2^).

SDNN and TP indicators reflect the total heart rate variability, an increase in RMSSD, pNN50%; HF reflects an increase in vagal influences on the heart rate; HR, SI, LF indicate sympathetic influences on the heart’s rhythm [21]. The VLF indicator is associated with the cerebral control mechanisms of cardiac activity indirectly through the afferent sensory neurons of the heart and, to a greater extent, reflects sympathetic activity [22].

The statistical data was processed using the Statistica software (StatSoft, Tulsa, OK, USA, v.13.0). The description of quantitative indicators is carried out with an indication of the Median, the range of values corresponding to the 25th and 75th percentiles (lower and upper quartiles). A comparison of quantitative variables in independent groups was determined by the Mann–Whitney U test (*p* < 0.05). Statistical cluster analysis was performed using hierarchical clustering (Ward’s method) and the k-means method. To conduct cluster analysis, z-transformation of the data was carried out to reduce the asymmetry in the distribution of variables. Correlation analysis was performed using Spearmen’s test (r) at *p* < 0.05.

## 3. Results

We identified two clusters, which differed in HRV parameters, IM duration, CIAS score and CIAS subscale scores. CIAS score and Wit, Com, and Tm subscale scores were higher (*p* = 0.015–0.041) in individuals included in cluster I than in individuals in cluster II (Table 1). 

Cluster I (*n* = 18) 27.8% (*n* = 5) included people with a stable IA pattern, 50% (*n* = 9) with a moderate risk of developing IA, and 22.2% (*n* = 4) with a minimal risk of developing IA. In cluster II (*n* =31), 67.7% (*n* = 21) of individuals were at a moderate risk of IA developing and 32.3% (*n* = 10) with a minimal risk of IA developing. IM time was longer in persons of cluster I when compared with individuals of cluster II (*p* = 0.041).

HRV indices also statistically differed between individuals included in cluster I and cluster II (Table 2).

RMSSD, SDNN, pNN50% and TP were statistically lower in cluster I individuals than in cluster II individuals (*p* = 0.001). A person of cluster I showed lower values of all components of the HRV spectrum (VLF, LF, HF, p = 0.003–0.001) and higher values of SI (*p* = 0.001) and HR (*p* = 0.001) when compared with individuals of cluster II. The range between minimum and maximum HR values in persons of cluster I was 65.9–108.2 bpm, while in persons of cluster II—65.8–102.9 bpm. In the individuals of cluster I, the Stress Index was more than 150 units, which indicates the predominance of sympathetic influences on the heart rhythm [23]. The individuals of cluster II had a Stress Index from 50 to 150 units, which indicates the balance of vagal and sympathetic regulatory influences on the heart rhythm. Thus, a decrease in vagal activity and a predominance of the sympathetic activity in the regulation of heart rhythm was revealed in individuals with reduced time control (cluster I). Individuals with preserved time control and less pronounced Internet-dependent behaviour (cluster II) had a higher total HRV, and a more balanced ratio of sympathetic and parasympathetic heart rate regulation.

A correlation analysis showed that IM value correlated with SI (r = 0.30, *p* = 0.033). Correlations were found between (Wit) score, on the one hand, and with HR (r = 0.36, *p* = 0.001), SI (r = 0.41, *p* = 0.003), SDNN (r = −0.31, *p* = 0.027), RMSSD (r = −0.37, *p* = 0.008), pNN50% (r = −0.37, *p* = 0.008), TP ms2 (r = −0.39, *p* = 0.004), HF ms2 (r = −0.35, *p* = 0.014) and LF (r = −0.44, *p* = 0.001), on the other hand.

## 4. Discussion

As a result of the cluster analysis, two clusters were identified, which differed the most in the studied parameters. Cluster I people had high values on CIAS and subscales of CIAS (Com, Wit, Tm), an extended time of an individual minute and low values of RMSSD, SDNN, pNN50%, TP, VLF, LF, HF against a background of high SI. Individuals of cluster II had average SIAS values, the time of an individual minute was within the normal range, the values of RMSSD, SDNN, pNN50%, TR were recorded higher, and SI values were lower than in individuals of cluster I. In adolescents of cluster I, which included people with different levels of IA risk and with a stable pattern of IA, the time for assessing an individual minute was more than 65 s. In addition, these people had significantly higher Tm, which reflects the subjective symptoms of time management disorders. Consequently, the subjective symptoms of a violation of time management in these people are confirmed by an objective method of control over time estimation. In cluster II, the assessment time was 61 s, which reflects a more accurate perception of time compared to individuals included in cluster I. The effectiveness of timing is highly dependent on various neural systems and is associated with cognitive functions such as attention and working memory [24]. Time perception ranging from seconds to minutes includes interneuronal interactions between the basal ganglia and the prefrontal cortex via dopaminergic and glutamatergic pathways [25,26]. The prefrontal cortex is part of the central autonomous network, which includes the insula, amygdala and the nucleus of the solitary tract [27]. A time perception is formed in the insular cortex, which is involved in the autonomic nervous control of visceral functions [28]. The accuracy of the timing can be functionally related to the autonomic nervous regulation of cardiac activity [29]. In our study, in individuals with reduced time control (cluster I), a decrease in vagal activity and a predominance of the activity of the sympathetic nervous system in the regulation of the heart rhythm were revealed. Individuals with preserved time control and a less pronounced risk of Internet-dependent behavior (cluster II) had a higher HRV, which reflects the balanced influence of the sympathetic and parasympathetic nervous systems in the regulation of the heart rhythm. HRV at rest is not only a reliable measure of the influence of the autonomic nervous system on heart function [3], but it also reflects the processes of the effectiveness of sustained attention [27]. High HRV at rest is associated with the processes of working memory and sustained attention, which may contribute to the allocation of attention resources required for effective timing [28]. It is assumed that low HRV is associated with a decrease in the activity of the prefrontal cortex. This state of the prefrontal cortex leads to disinhibition of the activity of the amygdala and sympathetic excitatory neurons in the rostral ventrolateral brain region of the medulla oblongata, as well as inhibition of parasympathetic excitatory neurons [30]. As a result of the above neurophysiological processes in the brain, there is an increase in heart rate and a concomitant decrease in HRV [30]. Low HRV is associated with difficulties in voluntary attention and inefficiency in the allocation of cognitive resources, which is reflected in the violation of subjective perception of time in the form of a significant lengthening of the time of an individual minute.

In our research, adolescents who were part of the cluster I, as opposed to the cluster II individuals, also have more pronounced withdrawal symptoms (Wit), which reflects a sense of discomfort in the absence of access to Internet resources [31], regardless of the presence or absence of Internet-addicted behavior. The correlations between the Wit score and HRV parameters reflect the tension of cardiac activity and a decrease in vagal reserves of autonomous regulation of the heart rate, in conjunction with an increase in the severity of withdrawal symptoms in the absence of access to Internet resources. Kim et al. (2016) demonstrated that long-term excessive internet gaming contributes to lower HRV levels in users, even at rest [32]. The authors suggested that internet gaming addiction may not reflect physical and/or psychological impairments in young adults, but they need to assess the risk of autonomic nerve dysregulation to avoid subsequent adverse effects on cardiac activity. We suggest that, in our study, increased values on the Wit subscale in combination with low HRV may be one of the risk factors for the development of Internet-addicted behavior in adolescents.

People with low HRV have a slow recovery of autonomous regulation of heart rate after cold exposure as the main environmental stress factor in the Arctic [33], as well as after psychological stress [34]. We also showed that the greatest severity of symptoms of withdrawal from access to the Internet was revealed in adolescents living in the northern region of Russia, in contrast to adolescents who are residents of the southern region of Russia [14]. In such conditions, low HRV, decreased time control and symptoms of discomfort in the absence of the Internet in adolescents reflect their difficulty in switching to other activities. At the same time, the risk of developing IA increases with the transition to the IA state, accompanied by a significant decrease in vagal reserves of autonomic regulation of the heart rate and the risk of cardiovascular disorders.

Thus, 16–17-year-old adolescents, who are residents of the Russian Arctic, with different levels of risk of developing Internet-dependent behaviour, including symptoms of a stable IA pattern, may have a high severity of symptoms of withdrawal from Internet use in conjunction with difficulty in time management, increased sympathetic activity and a relative decrease in vagal regulation of the heart rate. In individuals with minimal symptoms of Internet withdrawal, the total heart rate variability and vagal activity remain higher, and the time estimation remains more effective than in those with severe Internet withdrawal symptoms. The use of the Internet seems to be an important communication channel for adolescents in the uncomfortable environment of the Russian Arctic. Therefore, it is assumed that the withdrawal of access to Internet resources will be the leading symptom of Internet addiction associated with a disruption of the brain nerve networks that implement both time control and autonomic nervous regulation of cardiac activity.

To clarify the mechanisms of the risk of cardiac dysfunction during the formation of Internet-dependent behaviour, while taking into account the uncomfortable environment, it is important to continue similar studies in adolescents living in a comfortable climatic zone. 

The limitations of the study may be due to the specific climate environment, limited number, age, and gender of the study participants. Due to the mixed physiological significance of the “LF/HF ratio” (including both sympathetic and vagal influences on heart rate), this HRV parameter was not considered in this study.

## Figures and Tables

**Table 1 life-11-00497-t001:** CIAS subscale scores and IM duration in 16–17-year-old adolescents with different risks of IA ^a^.

Variables	Cluster I (*n* = 18)	Cluster II (*n* = 31)	*p*
CIAS, score	56.5 (44.0; 67.0)	47.0 (39.0; 52.0)	0.015
Com, score	11.0 (10.0; 12.0)	9.0 (7.0; 11.0)	0.022
Wit, score	12.0 (10.0; 14.0)	10.0 (8.0; 11.0)	0.001
Tol, score	9.5 (7.0; 12.0)	7.0 (6.0; 10.0)	0.066
IH-RP, score	11.5 (11.0; 15.0)	11.0 (9.0; 13.0)	0.219
Tm, score	11.5 (8.0; 14.0)	9.0 (7.0; 11.0)	0.041
IM, sec	67.0 (59.0; 74.0)	61.0 (59.0; 65.0)	0.041

CIAS: Chen Internet Addiction Scale; IM: individual minute; IA: internet addiction; Com: compulsive use; Wit: withdrawal symptoms, Tol: tolerance; IH-RP: interpersonal and health-related problems; Tm: time management problems. ^a^ Data are presented as Median (lower and upper quartiles).

**Table 2 life-11-00497-t002:** Heart rate variability parameters in 16–17-year-old adolescents with different risks of IA ^a^

Variables	Cluster I (*n* = 18)	Cluster II (*n* = 31)	*p*
HR, bpm	90.6 (85.9; 96.1)	79.3 (76.2; 86.4)	0.001
RMSSD, ms	18.9 (16.5; 24.1)	33.4 (27.1; 41.8)	0.001
SDNN, ms	32.2 (28.1; 36.3)	50.1 (40.6; 56.2)	0.001
pNN50, %	1.4 (0.7; 2.8)	13.9 (7.1; 25.9)	0.001
SI, units	311.1 (204.8; 417.1)	116.9 (87.9; 152.1)	0.001
TP, ms2	880.9 (674.8; 1249.9)	1898.8 (1558.9; 2721.3)	0.001
HF, ms2	204.7 (130.5; 335.9)	683.3 (433.4; 988.4)	0.001
LF, ms2	321.7 (229.3; 533.2)	637.7 (494.7; 870.2)	0.001
VLF, ms2	132.7 (98.7; 230.4)	237.7 (195.9; 310.3)	0.003

HR: heart rate; SDNN: standard deviation of all NN intervals; RMSSD: the square root of the mean squared differences of successive NN intervals; pNN50: percentage of the number of pairs of consecutive cardio intervals differing by more than 50 ms; SI: Stress Index; TP: total power of HRV spectrum; HF: high-frequency range of HRV spectrum; LF: low-frequency range of HRV spectrum; VLF: very-low-frequency range of HRV spectrum. ^a^ Data are presented as median (lower and upper quartiles).

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
