# Peer review of "Heart Rate Variability, Time Estimation and Internet-Dependent Behaviour in 16–17-Year-Old Adolescents: A Study in Russian Arctic"

_life, 2021, doi:10.3390/life11060497_

Round 1

Reviewer 1 Report

The present study by Krivonogova et al. shows different associations between Internet addiction, the effectiveness of time estimation, and a wide range of heart measures in adolescents living in the Russian Arctic. The results of a K-means cluster analysis revealed 2 clusters. The first cluster included people with different levels of IA risk and with a stable pattern of IA, the assessment of the individual minute was above 65 seconds. On the other hand, the second cluster included individuals with milder symptoms of Internet addiction and the assessment of the individual minute was more accurate than the individuals of the first cluster.  Regarding the heart, variability measures the results of the cluster analysis show that the subjects of cluster I exhibit lower values in RMSSD, SDNN, pNN50%, and TP, as well as, VLF, LF, HF, and higher levels of  SI and HR measures compared to the subjects of the second cluster.

The results  presented are novel and interesting but the minor concerns listed below should be addressed:

1) Seeing that the authors performed a cluster analysis I think that the objectives of the study should reflect that grouping similar individuals regarding the variables studied should be listed as one of the main objectives of this article.

2) The authors should clarify whether the 2 clusters solution was chosen based on previous literature or based on other methods such as the Silhouette plot or the Rand index.

3) Reliability measures of the test CIAS (Cronbach’s alpha) should be reported.

4) In the results I would suggest adding a table with the descriptive statistics for the whole sample

5) In line 146 in Table 1 legend TM should be Tm.

6) In the discussion a would suggest including a first paragraph summarizing the main differences between the clusters.

7) Considering that the heart variability measures are the ones where the differences between clusters are larger I would suggest that in the discussion the authors compared the present results with the ones of Huang et al., 2008 (reference number 4) or other studies such as Nahyun et al., 2016, https://doi.org/10.1089/cyber.2016.0282; or Hong et al., 2018, https://doi.org/10.3389/fpsyt.2018.00429.

8) In the reference list the authors of references  16, 20,22 are missing.

Author Response

We are very grateful to the referee for reviewing our article. 

Point 1: Seeing that the authors performed a cluster analysis I think that the objectives of the study should reflect that grouping similar individuals regarding the variables studied should be listed as one of the main objectives of this article.

Response 1. We have changed the research objective. Consequently, the aim of this study was to determine groups of individuals with different risk of IA in relation to HRV parameters and the efficiency of time estimation in adolescents living in the Russian Arctic.

Point 2. The authors should clarify whether the 2 clusters solution was chosen based on previous literature or based on other methods such as the Silhouette plot or the Rand index.

Response 2.

We've added information about clustering methods to the Materials and Methods section. We used two types of cluster analysis: hierarchical clustering (Ward's method) and k-means clustering. Hierarchical cluster analysis was necessary because the number of clusters was not known in advance. Dendrogram studies have shown that the two-cluster solution provides good differentiation between groups. The k-means cluster analysis was performed with two possible solutions. In the subsequent cluster analysis of k-means, we preferred hierarchical methods, since it is less sensitive to extreme values in the sample. As a consequence, more pronounced homogeneity within a cluster and differences between clusters were obtained.

Point 3. Reliability measures of the test CIAS (Cronbach’s alpha) should be reported.

Response 3.

We've added information in the Materials and Methods section. When studying Internet-dependent behavior in Russian people, the reliability measures of the test CIAS (Cronbach’s alpha) was ranged from 0.757 to 0.9, depending on the CIAS subscales [18, pp. 27-28]. In other words, CIAS test is presented with a sufficient degree of reliability.

Point 4. In the results I would suggest adding a table with the descriptive statistics for the whole sample

Response 4.

Since, according to the recommendation of the reviewer (Point 1), we clarified the purpose of the study, we considered it necessary to present the data in separate groups (clusters). We have clarified the significance levels between the parameters in the text.

Point 5. In line 146 in Table 1 legend TM should be Tm.

Response 5. We have made corrections to the legend of Table 1

Point 6. In the discussion a would suggest including a first paragraph summarizing the main differences between the clusters.

Response 6.

We've added the first paragraph to the Discussion section. As a result of the cluster analysis, 2 clusters were identified, which differed the most in the studied parameters. Cluster I people had high values on CIAS and subscales of CIAS (Com, Wit, Tm), an extended time of an individual minute and low values of RMSSD, SDNN, pNN50%, TP, VLF, LF, HF against a background of high SI. Individuals of cluster II had average SIAS values, the time of an individual minute was within the normal range, the values of RMSSD, SDNN, pNN50%, TR were recorded higher, and SI values were lower than in individuals of cluster I.

Point 7. Considering that the heart variability measures are the ones where the differences between clusters are larger I would suggest that in the discussion the authors compared the present results with the ones of Huang et al., 2008 (reference number 4) or other studies such as Nahyun et al., 2016, https://doi.org/10.1089/cyber.2016.0282; or Hong et al., 2018, https://doi.org/10.3389/fpsyt.2018.00429.

Response 7.

We compared our results with those of Nahyun Kim et al. (2016) and expanded the discussion.

In our research adolescents who were part of the cluster I, as opposed to the cluster II individuals, also have more pronounced withdrawal symptoms (Wit), which reflects a sense of discomfort in the absence of access to Internet resources, regardless of the presence or absence of Internet-related behavior. According to Nahyun Kim et al. (2016) demonstrated that long-term excessive internet gaming contributes to lower HRV levels in users, even at rest. The authors suggested that internet gaming addiction may not reflect physical and / or psychological impairments in young adults, but they need to assess the risk of autonomic nerve dysregulation to avoid subsequent adverse effects on cardiac activity. We suggest  that in our study, increased values on the Wit subscale in combination with low HRV may be one of the risk factors for the development of Internet-addicted behavior in adolescents.

Point 8. In the reference list the authors of references  16, 20,22 are missing.

Response 8.

Number 16 is a normative document of the Government of the Russian Federation; therefore, it does not contain specific authors.

Number 20 is a document - Task Force of the European Society of Cardiology the North American Society of Pacing Electrophysiology; therefore, it does not contain any authors.

Number 22 contains 12 authors, we have all included in the list.

Reviewer 2 Report

The authors studied HRV in 49 adolescents living in the arctic are of Russia (m:f=32:17; age 16-17 ys, 90% of them were in-country migrants from southern areas). 43% used the Internet more than 3h/d and were considered Internet addicts (IA, Chen Internet Addiction Scale >64/104). Assessments included individual time estimation (individual minute), HRV (mean, SD, square root of mean differences of successive NN intervals, the percentage of individual NN differences exceeding 50 msec, a stress index, and various other indexes, indicating activation of the sympathetic nervous system. Statistical calculations (median and range of upper and lower quartiles, and k-means clustering). Two clusters of HRV were compared with two clusters of IA (≥ vs. < 3h/d Internet activity), where HRV cluster 1 included 5/15 IA adolescents with IA, 9/19 with 2-3h/d Internet activity, and 4/15 with <2h/day Internet activity. The HRV values correlated with the daily Internet use, and the individual minute was longer in HRV cluster 1. The authors concluded that IA impairs the autonomic regulation because of increased stress in adolescents with IA.

The study is interesting because of increased stress may be related to future cardiovascular disease. There are, however, major problems including the study sample and the interpretation of results:

  1. Study sample:
    1. the number of investigated subjects is relatively low (45) if additional factors, internal migration, cold stress, m:f ratio (boys are more likely to be Internet addicted) are considered
    2. the sample is only insufficiently characterized: there are no data available on sleep disturbance, body mass index, psychosocial condition and acculturation stress, that may all influence the HR variability
    3. “stable” Internet addiction was defined as self-assessed Internet activity (gaming?) ≥ 3h/d. This is not an accepted definition. Mean (!) Internet activity, e.g. in German adolescents at present is 4.5 h/day.
  2. Interpretation of results:
    1. HRVCluster 1 is a “mixed” group containing only a third of  adolescents with “IA”. Therefore it is probable that other factors (e.g. sleep disturbance, internal migration and acculturation stress, climate) are responsible for the increased stress/diminished HRV despite the significance of differences between “IA” groups.
    2. The Discussion only insufficiently comments the results (e.g. the heterogeneity of HRV cluster 1, other explanations of stress load etc.), especially in view of the current literature, does not explain the relationship of increased sympathetic activity and slower time perception (that is not primarily expected) and does not discuss limitations of the study (small sample, more female than male participants, other causes of stress and decreased HRV,…)
  3. Minor comments:
    1. L 88: there are 15 adolescents with < 2 hrs IA!
    2. 59 & more: “stable pattern” of IA – data are cross sectional!
    3. L76: the participants were, l. 137: Spearman,
    4. 212 ff: withdrawal syndrome – better preoccupation with gaming / Internet activity

Author Response

We are very grateful to the referee for reviewing our article. Please see the attachment.

Point 1. The number of investigated subjects is relatively low (45) if additional factors, internal migration, cold stress, m:f ratio (boys are more likely to be Internet-addicted) are considered

Response 1.

We agree that the number of persons is not so large, since this is the first stage of the study, when studies are planned in different climatic-geographical regions of Russia. According to our data, there is no evidence yet that boys are more addicted to the Internet than girls. We will definitely check this hypothesis in the future. In this article, we added a section at the end of the discussion about research limitations. The limitations of the study may be due to the specific climate environment, limited number, age, and gender of the study participants.

Point 2. The sample is only insufficiently characterized: there are no data available on sleep disturbance, body mass index, psychosocial condition and acculturation stress, which may all influence the HR variability

Response 2.

We trust the data that the a risk of acculturation stress is significantly reduced in second-generation migrants born in their new homeland.  (Nicole E. Rossi, Toni L. Bisconti, C. S. Bergeman. (2007) The Role of Dispositional Resilience in Regaining Life Satisfaction after the Loss of a Spouse. Death Studies 31:10, pages 863-883). Since the parents of the study participants and the people living in the study region belonged to the same cultural-ethnic group, cross-cultural stress in both parents and study participants was considered insignificant.

We have added information that characterizes the individuals included in the sample. None of the participants smoked or consumed alcoholic drinks. Insomnia as a medical diagnosis was not verified in the study participants. The study participants had not committed any offences and did not need police or psychological supervision in connection with the manifestation of aggression, depression, and suicidal behaviour or the use of psychoactive substances. Anthropometric parameters (height and body mass index [BMI]) were measured for each participant in the school medical office. BMI was defined as the body mass divided by the square of the body height (kg/m2), with mass in kilograms and height in meters. Height, weight, and BMI were measured using certified medical equipment with electronic scales and a height meter ("REP+VMEN-200–100-D1-A", Russian Federation). The adolescents were not underweight or overweight (BMI 18.5-24.9 kg /m2).

Point 3. “Stable” Internet addiction was defined as self-assessed Internet activity (gaming?) ≥ 3h/d. This is not an accepted definition. Mean (!) Internet activity, e.g. in German adolescents at present is 4.5 h/day.

Response 3.

We confirm that the "stable" state of internet addiction is regarded as a subjective state according to the Chen test. We have provided an explanation of the times used Internet recourses by adolescents. The total time for using the Internet (for the purpose of  learning in the school curriculum and for entertainment) averaged 2.5 hours for 8.5% of individuals, 5.5 hours for 42.9% of individuals, and 6 or more hours for 48.6% of individuals every day. The entertainment time included social media time, watching video content and news, and playing online. For the purpose of entertainment, 15 people (42.9%) used the Internet 3-4 hours a day, 19 people (54.3%) used the Internet 2-3 hours a day and 1 person (2.8 %) used the Internet up to 2 hours a day.

Point 4. HRVCluster 1 is a “mixed” group containing only a third of  adolescents with “IA”. Therefore it is probable that other factors (e.g. sleep disturbance, internal migration and acculturation stress, climate) are responsible for the increased stress/diminished HRV despite the significance of differences between “IA” groups.

Response 4.

We suggest that even a high risk of Internet addiction (without signs of a «stable» pattern of IA), expressed in higher СIAS scores and more pronounced individual symptoms of it, reflect the risk of impaired autonomic regulation of heart rhythm and time perception, which we try to demonstrate in the article. Although we agree that in other social and climatic conditions, the dependence of HRV, estimates of time on the severity of Internet addictive behavior will differ from the presented data. We believe that clearer data on the influence of socio-demographic factors on the relationship between the risk of Internet-addicted behavior and its manifestation, HRV and time estimates will be obtained in a comparative analysis with the studied indicators of people living in the southern regions of Russia. That is why it is important for us to show such patterns in specific climatic conditions.

Point 5.The Discussion only insufficiently comments the results (e.g. the heterogeneity of HRV cluster 1, other explanations of stress load etc.), especially in view of the current literature, does not explain the relationship of increased sympathetic activity and slower time perception (that is not primarily expected) and does not discuss limitations of the study (small sample, more female than male participants, other causes of stress and decreased HRV,…)

Response 5.

This study is part of a larger neurophysiological and psychological study that includes a number of indicators not included in this article. The total research time per person is 55-60 minutes. At a given time, under the conditions of an expedition to the Arctic region, we were able to examine just such a number of people using such a program. Girls were more willing to participate in the survey, while boys preferred to avoid and hide information about their health.

We've added information to the Discussion section.

  • As a result of the cluster analysis, 2 clusters were identified, which differed the most in the studied parameters. Cluster I people had high values on CIAS and subscales of CIAS (Com, Wit, Tm), an extended time of an individual minute and low values of RMSSD, SDNN, pNN50%, TP, VLF, LF, HF against a background of high SI. Individuals of cluster II had average SIAS values, the time of an individual minute was within the normal range, the values of RMSSD, SDNN, pNN50%, TR were recorded higher, and SI values were lower than in individuals of cluster I.

2) In our study, in individuals with reduced time control (cluster I), a decrease in vagal activity and a predominance of the activity of the sympathetic nervous system in the regulation of the heart rhythm was revealed. Individuals with preserved time control and a less pronounced risk of Internet-dependent behavior (cluster II) had a higher HRV, which reflects the balanced influence of the sympathetic and parasympathetic nervous systems in the regulation of the heart rhythm.

  • As a result of the above neurophysiological processes in the brain, there is an increase in heart rate and a concomitant decrease in HRV [Thayer JF, Lane RD.2009 [29].Low HRV is associated with difficulties in voluntary attention and inefficiency in the allocation of cognitive resources, which is reflected in the violation of subjective perception of time in the form of a significant lengthening of the time of an individual minute.
  • The limitations of the study may be due to the specific climate environment, limited number, age, and gender of the study participants.

Point 6. Minor comments:

    1. L 88: there are 15 adolescents with < 2 hrs IA!
    2. 59 & more: “stable pattern” of IA – data are cross sectional!
    3. L76: the participants were, l. 137: Spearman,
    4. 212 ff: withdrawal syndrome – better preoccupation with gaming / Internet activity

Response 6.

We are grateful for the minor comments, but we tried to explain them in the relevant paragraphs (Points 1-5).

Reviewer 3 Report

In my opinion, the manuscript requires the following additions:
1. The introduction must be better written so that it introduces the subject more broadly. Information on the influence of other behavioral factors (e.g. stress) and environmental factors (e.g. environmental tobacco smoke, cell phone electromagnetic fields) on heart rate variability must be added.
2. Table 1 presenting the basic characteristics of the study group (age, height, weight, body mass index, percentage of overweight and obese people, gender distribution, race distribution) should be added.
3. The measurement of the HR parameter must be explained. It would be optimal to include the minimum HR, average HR and maximum HR in the results.
4. The HRV analysis should be supplemented with the LF: HF ratio, which is an indicator of the autonomic balance.
5. The paragraph on the limitations of the study should be added to the discussion chapter. The small size of the study group and other limitations of the study need to be mentioned and discussed.

Author Response

We are very grateful to the referee for reviewing our article. Please see the attachment.

Point 1. The introduction must be better written so that it introduces the subject more broadly. Information on the influence of other behavioral factors (e.g. stress) and environmental factors (e.g. environmental tobacco smoke, cell phone electromagnetic fields) on heart rate variability must be added.

Response 1.

We have added information to the Introduction section.

A healthy person has a high HRV, which reflects a readiness to respond to environmental stimuli. A decrease in HRV indicates the possibility of pathological conditions, including cardiovascular diseases and is detected in people who consume alcohol, tobacco, with prolonged exposure to electromagnetic fields, with fatigue, depression, anxiety, with the experience of negative emotions (Fatisson J, Oswald V, Lalonde F., 2016)

Point 2. Table 1 presenting the basic characteristics of the study group (age, height, weight, body mass index, percentage of overweight and obese people, gender distribution, race distribution) should be added.

Response 2.

We have added information to the Materials and Methods section about socio-demographic characteristics, body mass index and ethnicity of study participants.

All study participants were Europeoid residents. None of the participants smoked or consumed alcoholic drinks. Insomnia as a medical diagnosis was not verified in the study participants. The study participants had not committed any offences and did not need police or psychological supervision in connection with the manifestation of aggression, depression, and suicidal behaviour or the use of psychoactive substances. Anthropometric parameters (height and body mass index [BMI]) were measured for each participant in the school medical office. BMI was defined as the body mass divided by the square of the body height (kg/m2), with mass in kilograms and height in meters. Height, weight, and BMI were measured using certified medical equipment with electronic scales and a height meter ("REP+VMEN-200–100-D1-A", Russian Federation). The adolescents were not underweight or overweight (BMI 18.5-24.9 kg /m2).

Point 3. The measurement of the HR parameter must be explained. It would be optimal to include the minimum HR, average HR and maximum HR in the results.

Response 3.

We have added information to the Materials and Methods section. HRV indices were recorded in a person in a sitting position for 5 minutes. To characterize the heart rate (HR), we used the average heart rate for 5 minutes of HRV recording.

Point 4. The HRV analysis should be supplemented with the LF: HF ratio, which is an indicator of the autonomic balance.

Response 4.

Many authors have different opinions on the interpretation of the LF indicator, which may be associated not with sympathetic activity, but with baroreflex activity (Fatisson J, Oswald V, Lalonde F., 2016).Therefore, we avoided analyzing the LF /HF ratio to describe the activity of the autonomic nervous system divisions.

Point 5. The paragraph on the limitations of the study should be added to the discussion chapter. The small size of the study group and other limitations of the study need to be mentioned and discussed.

Response 5.

We have added information about the limitations of the study.

The limitations of the study may be due to the specific climate environment, limited number, age, and gender of the study participants.

Round 2

Reviewer 3 Report

Most of my suggestions have been incorporated into the new version of the manuscript. 2 more comments:
1. The variables "minimum HR" and "maximum HR" must be added to the analysis.
2. The missing "LF / HF ratio" should be listed as a study limitation.

Author Response

We are grateful to the Reviewer for the great work of reviewing our article, which made our stady better.

Comment 1: The variables "minimum HR" and "maximum HR" must be added to the analysis.

Response 1: The range between minimum and maximum HR values in persons of cluster I was 65,9-108,2 bpm, while in persons of cluster II - 65,8-102,9 bpm. This data has been added to the text of this article (3.Results).

Comment 2: The missing "LF / HF ratio" should be listed as a study limitation.

Response 2: Due to the mixed physiological significance of the “LF/HF ratio” (including both sympathetic and vagal influences on heart rate), this HRV parameter was not considered in this study. This data has been added to the text of this article (The limitations of the study).